# Opportunistic screening for type 2 diabetes in community pharmacies. Results from a region-wide experience in Italy

**Roberto Gnavi**[1]*, **Veronica Sciannameo**[1], **Francesca Baratta**[2], **Cecilia Scarinzi**[1], **Marco Parente**[2], **Massimo Mana**[3], **Mario Giaccone**[4], **Paolo Cavallo Perin**[5], **Giuseppe Costa**[1,6], **Teresa Spadea**[1], **Paola Brusa**[2]

**1** Epidemiology Unit, ASL TO3, Grugliasco (TO), Italy, **2** Department of Drug Science and Technology, University of Torino, Torino TO, Italy, **3** ATF Informatics, Cuneo, Italy, **4** Order of Pharmacists of Torino, Torino TO, Italy, **5** Department of Medical Sciences, University of Torino, Torino TO, Italy, **6** Department of Clinical and Biological Sciences, University of Torino, Torino TO, Italy

* roberto.gnavi@epi.piemonte.it

**Data Availability Statement:** URL.https://zenodo.org/ doi:10.5281/zenodo.3648088.

## Abstract

### Background and aims

Given the paucity of symptoms in the early stages of type 2 diabetes, its diagnosis is often made when complications have already arisen. Although systematic population-based screening is not recommended, there is room to experience new strategies for improving early diagnosis of the disease in high risk subjects. We report the results of an opportunistic screening for diabetes, implemented in the setting of community pharmacies.

### Methods and results

To identify people at high risk to develop diabetes, pharmacists were trained to administer FINDRISC questionnaire to overweight, diabetes-free customers aged 45 or more. Each interviewee was followed for 365 days, searching in the administrative database whether he/she had a glycaemic or HbA1c test, or a diabetologists consultation, and to detect any new diagnosis of diabetes defined by either a prescription of any anti-hyperglycaemic drug, or the enrolment in the register of patients, or a hospital discharge with a diagnosis of diabetes.

Out of 5977 interviewees, 53% were at risk of developing diabetes. An elevated FINDRISC score was associated with higher age, lower education, and living alone. Excluding the number of cases expected, based on the incidence rate of diabetes in the population, 51 new cases were identified, one every 117 interviews. FINDRISC score, being a male and living alone were significantly associated with the diagnosis.

### Conclusions

The implementation of a community pharmacy-based screening programme can contribute to reduce the burden of the disease, particularly focusing on people at higher risk, such as the elderly and the socially vulnerable.

**Funding:** The study has been supported by a financial grant from the Italian Ministry of Health, and an unconditional grant from Federfarma Piemonte. Federfarma is a syndicate (representative of pharmacists), not a company with commercial purposes. Federfarma only provided financial support in the form of research materials (such as questionnaires), and did not have any additional role in the management of the study.

**Competing interests:** The authors have declared that no competing interests exist.

# Introduction

It is well known that the number of people with diabetes and its prevalence are increasing worldwide. In Italy, diabetes affects about 3.4 million of people [1], and, due to its burden in terms of social and health costs, it represents an important public health issue.

Despite the increased awareness of the disease in the last years, a considerable number of diagnosis is still made when severe complications, both micro and macrovascular, have already arisen [2]; furthermore, the delayed diagnosis is more frequent among people from low socio-economic status [3]. This implies that, even if the efficacy of systematic population-based screening is still under debate and not recommended by some health institutions [4,5], there is room to experience new strategies for improving early diagnosis of type 2 diabetes, whose symptoms become apparent several years after the onset of the disease. Opportunistic screening is recommended for high risk individuals, as early detection of the disease enables to initiate therapies aimed to improve glycaemic control and, consequently, to reduce or delay the onset of complications [2].

The Finnish Diabetes Risk Score (FINDRISC) questionnaire is a commonly used scorecard that predicts the probability of developing type 2 diabetes over the following 10 years [6], thus allowing identifying patients at high risk of having undiagnosed diabetes. These individuals can be addressed to a second level test such as measurement of fasting plasma glucose or HbA1c concentration to confirm or rule out the diagnosis of diabetes. FINDRISC has been validated in Italy [7].

Public pharmacies, being very accessible, are frequently a contact point with the health care system [8]. In Italy there are more than 19000 community pharmacies, 1500 in Piedmont. As they are distributed throughout the country (nearly every municipality has at least one pharmacy), easy to access and free of charge, they are considered by the population as a fast and trustworthy gateway to health services. Therefore, pharmacies are strategically placed to reach out large part of the population. It is against this background that, in 2012, the Regional Orders of Pharmacists, Federfarma Piemonte and the University of Torino have launched an extensive programme aimed to counteract the negative effects of diabetes. The programme was based upon two main actions: identification of undiagnosed cases of the disease among customers of community pharmacies by filling out the Italian FINDRISC questionnaire [9], and increasing adherence to guidelines among people with confirmed diabetes. Subsequently, in 2016, the Italian Health Ministry has funded a study to assess the efficacy of the programme.

We report the results of the opportunistic screening of type 2 diabetes in terms of people at high risk identified, and new cases of confirmed diabetes detected. We discuss the implications of these results in terms of public health.

# Materials and methods

## Study population and protocol of the intervention

The intervention has been conducted in Piedmont, 4.4 million inhabitants in North West Italy, in two steps: first, as a cross-sectional survey to identify people at high risk to develop diabetes; second as a follow up study of all the interviewees, to identify those who developed diabetes (Azienda Sanitaria Locale A.S.L. TO2 Ethical Committee Approval Protocol n˚46480/2013).

All pharmacists operating in private and public community pharmacies of Piedmont were invited to participate on a voluntary basis, and without any payment, to the project. Those who agreed were enrolled in a training course on diabetes (conducted by senior

diabetologists), and on the study procedures and instruments, with special attention to the items of the FINDRISC questionnaire, to ensure that all pharmacists collected data homogeneously.

FINDRISC includes eight questions concerning well established risk factor for type 2 diabetes: age, body mass index, waist circumference, physical activity, daily consumption of vegetables, history of antihypertensive drugs, family history of diabetes, and history of fasting plasma glucose. Based on the answers to these questions, it generates a score ranging from 0 to 26. The score is categorized in four classes, representing the probability of developing diabetes within the following 10 years: <12 (low risk, ≤4%), 12–14 (moderate, 5–17%), 15–20 (high, up to 33%), >20 (very high, ≥50%).

During the two periods October 2013–March 2014 and November 2014–April 2015, all subjects aged 45 or more entering a pharmacy, and looking overweight (based on the pharmacist judgement), or with a family history of diabetes (presenting a prescription for any antidiabetic drug for a close relative), were invited to participate. Those who agreed, gave their written informed consent to be interviewed and followed-up. Individuals that reported to be affected with diabetes were excluded.

Participants were interviewed by the pharmacists in a consultation room within each pharmacy, and had their height, weight (in light clothes and without shoes) and waist circumference (using an unextendable measuring band) measured. All data were electronically stored in a central database.

In addition to the questions included in the FINDRISC, the questionnaire also included information on educational level and household condition. Educational level, measured as the maximum attained degree, was categorized in five classes: no formal education (corresponding to International Standard Classification of Education 1997(ISCED97 level 0), primary school (ISCED97 level 1), middle school (ISCED97 levels 2-3C), high school (ISCED97 levels 3A-3B), and university degrees (ISCED97 levels 5–6) [10]. Household condition was represented by a dichotomous variable indicating if the patient lived alone or not.

Age at enrolment was categorized in three classes (same as FINDRISC): 45–54 years, 55–64 years and more than 64 years.

At the end of the interview all individuals at risk received counselling on physical activity and correct diet, and, in case of a FINDRISC higher than 12, were referred to their general practitioner for possible further examinations.

## Outcome definition and recording

The whole population of Piedmont is covered by an automated system of databases recording individual data on hospitalizations, outpatient healthcare services, drugs dispensed from pharmacies reimbursed by the National Health System, exemptions from co-payment of drugs and laboratory examinations (mainly because of chronic diseases). All these archives can be linked together by a unique anonymous identifier that is encrypted to protect the privacy of patients. Furthermore, all these data are linked to the regional population register to follow population in case of mortality or moving outside of Piedmont.

The database of the interviewees, specifically created for this protocol, was anonymized using the same encryption algorithm used to encrypt all regional health databases. Thus, it was possible to link the databases one to the other anonymously. After the linkage, interviewees who resulted to be not diabetes free (because prescribed with an anti-hyperglycaemic drug before the interview, or previously discharged from hospital with a diagnosis of diabetes, or exempted from co-payment because of diabetes), were excluded. In case of more than one interview for the same individual from different pharmacies, only the first one was retained.

Subsequently, each individual of this diabetes-free population was followed for 365 days after the interview date, searching in the administrative database whether they had a glycaemic or HbA1c test, or a diabetologist consultation. Finally, we defined as a new case of diabetes either a prescription of any anti-hyperglycaemic drug (either oral ATC = A10B, or insulin ATC = A10A), or the enrolment in the register of patients exempted from co-payment because of a confirmed diagnosis of diabetes, or a discharge from hospital with a diagnosis of diabetes (ICD9CM 250).

Since the programme was launched simultaneously throughout the region, we could not introduce a pharmacy-based control group. Therefore, we used historical data to estimate the incidence of diabetes in the absence of screening and calculate the additional number of newly diagnosed cases due to the screening process. There are no national data on the incidence of type 2 diabetes in Italy, but only a few local studies, which report rates ranging from 5 to 8 cases x 1000 person-years [11,12], while in UK the incidence was estimated to be 7.3 x 1000py [13]; we hence used a conservative estimate of a 7x1000 incidence rate.

## Statistical analysis

Quantitative variables were described using means and standard deviations (SD); qualitative data were described via frequencies and percentages. To assess differences we used Kruskal-Wallis tests for continuous variables and Chi squared tests for categorical variables; a 2-tail p-value less than 0.05 was considered statistically significant.

Determinants of high FINDRISC score and of diabetes diagnosis were investigated performing log-binomial univariate and multivariate regression models, estimating prevalence rate ratio (PRR) with their 95% confidence intervals (95% CI).

All statistical analyses were performed using SAS-ver.9.3.

## Results

In total, 1400 pharmacies (93% of the total number of pharmacies in Piedmont) took part in the training course with at least one pharmacist. Of these, 855 participated in the regional programme entailing both opportunistic screening and adherence to treatment; 443 pharmacies were assigned to opportunistic screening. Pharmacists conducted 7024 interviews. Subsequent quality controls on the database led to the exclusion of 338 duplicated questionnaires (4.8%), 533 subjects who had already been diagnosed with diabetes (7.6%), and 125 aged less than 45 years (1.8%). Finally, for the purposes of this study, we also excluded 51 individuals who died or moved out of Piedmont during the follow-up period. The final study population consisted of 5977 diabetes-free individuals, who were still alive and resident in Piedmont one year after the interview (Fig 1).

Their FINDRISC score and sociodemographic characteristics are shown in Table 1. The FINDRISC score was low (<12) only for 47% of individuals; as for the remaining subjects, 23.9% had a moderate risk of developing diabetes in the following 10 years, 25.5% a high risk, and 3.4% a very high risk. The groups at higher risk were older, had higher prevalence of overweight/obesity, hypertension, hyperglycaemia, and lower levels of physical inactivity. Moreover, people at higher risk were more likely to have low educational levels, while there were no gender differences.

During the 12 months following the interview, 47.8% of interviewees measured their plasma glucose level, 12.1% HbA1c, and 5.2% were seen by a diabetologist. Among people at moderate, high or very high risk (FINDRISC > = 12) these percentages overall were 53.3, 17.6 and 6.7, respectively; as expected, the frequencies increased with increasing FINDRISC score (Table 2).

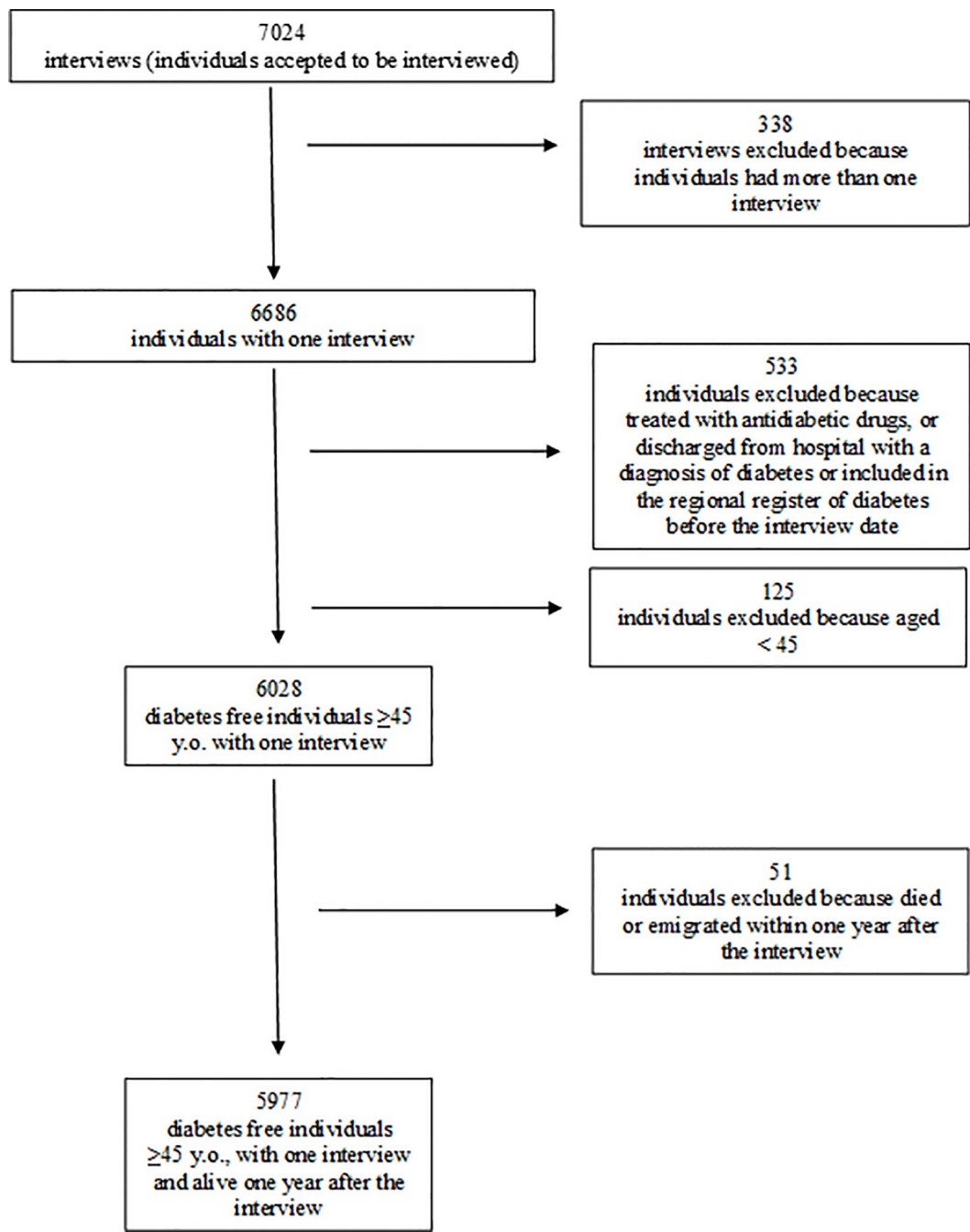

**Fig 1. Selection of the study population.**

Finally, a total of 107 individuals had a confirmed diagnosis of diabetes, corresponding to 1.8% out of 5977 interviewees. Among those with FINDRISC > = 12, 93 new cases of diabetes were detected, ranging from 2.0% among individuals at moderate risk to 8.4% in those at very high risk. It should be noted that 14 cases of diabetes were diagnosed among those who did not result at risk (i.e. FINDRISC< 12). To correctly calculate the real number of new cases diagnosed thanks to the screening process, we removed the cases that would be diagnosed even if no screening were active. Thus, applying the rate of 7x1000 on the study population of

**Table 1. Interviewed people by socio-demographic, FINDRISC characteristics and FINDRISC score.**

| | Total (n = 5977) | | FINDRISC<12 (n = 2820) | | FINDRISC 12–14 (n = 1430) | | FINDRISC 15–20 (n = 1525) | | FINDRISC>20 (n = 202) | | p |
|---|---|---|---|---|---|---|---|---|---|---|---|
| | n | % | n | % | n | % | n | % | n | % | |
| **Sex** | | | | | | | | | | | 0.47 |
| Men | 2286 | 38.3 | 1070 | 37.9 | 531 | 37.1 | 602 | 39.5 | 83 | 41.1 | |
| Women | 3691 | 61.8 | 1750 | 62.1 | 899 | 62.9 | 923 | 60.5 | 119 | 58.9 | |
| **FINDRISC** | | | | | | | | | | | |
| Age (mean and SD) | 63.3 | 10.8 | 61.4 | 10.9 | 64.0 | 10.6 | 65.7 | 10.2 | 67.6 | 9.8 | <0.001 |
| 45–54 | 1519 | 25.4 | 929 | 32.9 | 329 | 23.0 | 240 | 15.7 | 21 | 10.4 | <0.001 |
| 55–64 | 1740 | 29.1 | 839 | 30.0 | 420 | 29.4 | 429 | 28.1 | 52 | 25.7 | |
| >64 | 2718 | 45.5 | 1052 | 37.3 | 681 | 47.6 | 856 | 56.1 | 129 | 63.9 | |
| BMI (mean) | 26.1 | 4.5 | 24.0 | 3.4 | 26.8 | 4.0 | 28.7 | 4.5 | 30.4 | 4.6 | <0.001 |
| Waist circ. (mean) | 93.8 | 14.2 | 87.4 | 12.4 | 96.5 | 12.5 | 101.6 | 12.9 | 106.4 | 14.9 | <0.001 |
| Low physical activity[a] | 3158 | 52.8 | 1868 | 66.2 | 697 | 48.7 | 546 | 35.8 | 47 | 23.3 | <0.001 |
| Low veg. consumption[b] | 5056 | 84.6 | 2472 | 87.7 | 1187 | 83.0 | 1243 | 81.5 | 154 | 76.2 | <0.001 |
| Anti-hypertensive medication | 2776 | 46.4 | 815 | 28.9 | 736 | 51.5 | 1048 | 68.7 | 177 | 87.6 | <0.001 |
| Relatives diagnosed with diabetes[c] | 2965 | 49.6 | 797 | 28.3 | 792 | 55.4 | 1176 | 77.1 | 200 | 99.0 | <0.001 |
| High glycemic levels[d] | 1075 | 18.0 | 73 | 2.6 | 198 | 13.9 | 605 | 39.7 | 199 | 98.5 | <0.001 |
| **Educational level** | | | | | | | | | | | <0.001 |
| None | 314 | 5.3 | 122 | 4.3 | 78 | 5.4 | 96 | 6.3 | 18 | 8.9 | |
| Primary | 1182 | 19.8 | 418 | 14.8 | 311 | 21.8 | 391 | 25.6 | 62 | 30.7 | |
| Middle | 2341 | 39.2 | 1043 | 37.0 | 575 | 40.2 | 647 | 42.4 | 76 | 37.6 | |
| High | 1433 | 24.0 | 795 | 28.2 | 336 | 23.5 | 274 | 18.0 | 28 | 13.9 | |
| University | 707 | 11.8 | 442 | 15.7 | 130 | 9.1 | 117 | 7.7 | 18 | 8.9 | |
| **Household** | | | | | | | | | | | 0.05 |
| Living alone | 327 | 5.5 | 130 | 4.6 | 89 | 6.2 | 95 | 6.2 | 13 | 6.4 | |
| Not living alone | 5650 | 94.5 | 2690 | 95.4 | 1341 | 93.8 | 1430 | 93.8 | 189 | 93.6 | |

[a] daily physical activity at work or during leisure time < 30 minutes.

[b] vegetable, fruit, or berries not every day.

[c] any of the members of the immediate family or other relatives diagnosed with diabetes.

[d] ever been found to have high blood glucose.

5977 subjects, we would expect a maximum of 42 incident cases of type 2 diabetes in one year, in the absence of screening. Therefore, the real number of new cases detected would amount to 51 (=93–42), which is 0.85% of the population meeting the enrolment criteria. In other words, pharmacists should interview 117 (=5977/51) correctly selected customers (i.e.

**Table 2. Ascertainment of diabetes by FINDRISC score.**

| | FINDRISC<12 (n = 2820) | | FINDRISC 12–14 (n = 1430) | | FINDRISC 15–20 (n = 1525) | | FINDRISC>20 (n = 202) | | p |
|---|---|---|---|---|---|---|---|---|---|
| | n | % | n | % | n | % | n | % | |
| Glycemia | 1175 | 41.7 | 679 | 47.5 | 861 | 56.5 | 142 | 70.3 | <0.0001 |
| HbA1c | 165 | 5.9 | 177 | 12.4 | 305 | 20.0 | 74 | 36.6 | <0.0001 |
| Diabetogist consultation | 96 | 3.4 | 53 | 3.7 | 131 | 8.6 | 28 | 13.9 | <0.0001 |
| Confirmed diabetes | 14 | 0.5 | 28 | 2.0 | 48 | 3.2 | 17 | 8.4 | <0.0001 |

**Table 3. Determinants of moderate/high FINDRISC score and of diabetes diagnosis.**

| | FINDRISC > = 12 | | | | | Diagnosed diabetes | | | | |
| | (n = 3157) | | | | | (n = 93) | | | | |
| | n | PRR | 95% CI | Adj PRR | 95% CI | n | PRR | 95% CI | Adj PRR | 95% CI |
|---|---|---|---|---|---|---|---|---|---|---|
| **Sex** | | | | | | | | | | |
| Men | 1216 | 1 | | 1 | | 45 | 1 | | 1 | |
| Women | 1941 | 0.99 | 0.94–1.04 | 1.00 | 0.96–1.05 | 48 | 0.67 | 0.45–0.99 | 0.65 | 0.44–0.98 |
| **Age** | | | | | | | | | | |
| 45–54 | 590 | 1 | | 1 | | 10 | 1 | | 1 | |
| 55–64 | 901 | 1.33 | 1.23–1.44 | 1.31 | 1.21–1.41 | 23 | 1.51 | 0.72–3.14 | 1.33 | 0.64–2.78 |
| >64 | 1666 | 1.58 | 1.47–1.69 | 1.43 | 1.33–1.54 | 60 | 2.12 | 1.09–4.12 | 1.55 | 0.78–3.11 |
| **Educational level** | | | | | | | | | | |
| High/Degree | 903 | 1 | | 1 | | 25 | 1 | | 1 | |
| Middle | 1298 | 1.31 | 1.24–1.40 | 1.26 | 1.19–1.34 | 29 | 0.80 | 0.48–1.37 | 0.77 | 0.45–1.31 |
| None/Elementary | 956 | 1.51 | 1.42–1.61 | 1.34 | 1.26–1.44 | 39 | 1.47 | 0.90–2.41 | 1.22 | 0.72–2.07 |
| **Living in a household** | | | | | | | | | | |
| Not alone | 2960 | 1 | | 1 | | 82 | 1 | | 1 | |
| Alone | 197 | 1.15 | 1.05–1.26 | 1.07 | 0.98–1.17 | 11 | 2.02 | 1.09–3.72 | 1.97 | 1.07–3.63 |
| **FINDRISC** | | | | | | | | | | |
| 12–14 | | | | | | 28 | 1 | | 1 | |
| 15–20 | | | | | | 48 | 1.61 | 1.01–2.55 | 1.54 | 0.97–2.44 |
| >20 | | | | | | 17 | 4.30 | 2.40–7.71 | 3.93 | 2.18–7.05 |

diabetes-free individuals, aged 45 or more, and overweight or with family history of diabetes) to detect one new case of diabetes.

Table 3 reports the results of two multivariate models analysing determinants of getting a high FINDRISC score, i.e. being considered at risk (> = 12) and of being diagnosed with diabetes among individuals at risk. An elevated score was positively associated with lower educational levels, and living alone. As for newly diagnosed diabetes, after adjusting for FINDRISC score, being a male and living alone were the only determinants showing a statistically significant association with the diagnosis; age and educational level showed the same pattern as in the model for a high FINDRISC, but they were not statistically significant.

## Discussion

The first conclusion that can be drawn from our study is that an opportunistic screening for diabetes in the community pharmacy setting, using a validated tool as the FINDRISC, is feasible, and can detect both people at high risk of developing diabetes and previously unknown cases of the disease. Males, the elderly, low educated people and those living alone are those who benefit more.

Few experiences of pharmacy-based opportunistic screening for diabetes have been published in different countries such as Australia [14], Switzerland [15], Spain [16,17], each of them based on different questionnaires and scorecards. All of them reported that pharmacies, also thanks to their high accessibility, are an appropriate setting for screening for diabetes, as well as for other risk factors for cardiovascular diseases [18,19]. Only one pharmacy-based screening for persons at risk of diabetes has been carried out in Italy, using the FINDRISC questionnaire as screening tool [20]. Differently from our study, their target population included any subject aged 18 or more, without any selection based on overweight or family history of diabetes. Consequently, they found a lower prevalence of high-risk subjects compared

to our findings (34% vs 52%). All these studies concluded their screening process by referring high-risk individuals to their general practitioner for further evaluation; none assessed the efficacy of the screening in terms of new cases of diabetes detected.

In our study, more than half of the interviewees were considered at risk of developing diabetes; only half of them had their glucose levels measured, leading to 93 new diagnosis of diabetes during 1 year of follow-up. It must be noted that this should be regarded as the "minimum" number of confirmed cases, since further new cases are likely to be diagnosed, but not yet tracked by information systems. We also showed that, excluding the number of incident cases that would "spontaneously" arise in the study population in the absence of the opportunistic screening, pharmacists should interview 117 customers to detect one new case of type 2 diabetes. Again, this is a conservative estimate, since we used a 7x1000 rate, which is consistent with the highest rates estimated for Italy [11,12], and for UK [13]. Comparisons with similar data available in the literature are difficult, or even impossible, because of different diseases object of screening, different characteristics of the screened populations, and different methodology to calculate the number needed to be screened [21,22]. Therefore, the above mentioned ratio is a new element that decision-makers should take into account when carrying out an evaluation of the implementation of a community pharmacy-based programme to detect undiagnosed cases of diabetes.

We believe that there is space to improve this ratio, reducing the number of interviews needed to detect one case. First, focusing on subjects who get higher FINDRISC scores, as the elderly, the overweight and the obese, would improve the efficiency of the opportunistic screening. Indeed, in the aforementioned Italian study that enrolled subjects without any selection, only one third of the interviewed subjects had a FINDRISC score above 12 (compared to more than half in our study) [20]. Secondly, since only half of subjects at risk had measured their blood glucose levels, the screening programme should pay more attention to pharmacist-patient communication, and above all it should improve the involvement of general practitioners, so that more people classified as "at risk" may receive the appropriate second level diagnostic tests.

One further element to take into account is that the prevalence of subjects scoring 12 or more at FINDRISC is higher among lower educated people, and that new diagnoses are more frequent in those living alone. Unhealthy, diabetes-related, behaviours such as smoking, overweight/obesity, physical inactivity are more common among low educated people compared to high educated [23]; consequently, both incidence and prevalence of diabetes are more frequent among the more disadvantaged [24,25]. Therefore, as also undiagnosed diabetes is more frequent within these strata of the population [3], our results suggest a possible moderating impact of the programme on socioeconomic inequities, similarly to what reported for cancer screening [26,27].

The main strength of our work is that, to the best of our knowledge, it is the first study to evaluate the feasibility of a pharmacy-based screening for diabetes by following all screened individuals for one year after the interview. Other studies show that people at high risk can be successfully detected in pharmacy and referred to their practitioners, but they do not show if detected patients eventually result in a confirmed diagnosis [19].

Our study has limitations that could affect the results. First, the voluntary participation of pharmacists to the study has likely selected self-motivated professionals that could have collected replies to the questionnaires more carefully. Secondly, and partially consequent to the previous limitation, the sample of participants is unlikely representative of the whole population of Piedmont, and, even less, of Italy. These two points imply that, in case of a wider implementation of this screening programme, the same results cannot be ensured, unless an appropriate monitoring system of the extended programme is implemented.

Furthermore, our results are based on data coming from administrative information systems, which only record medical services reimbursed by the regional health system: individuals could have had their blood glucose levels measured privately, as well as their medical consultations, thus resulting in an underestimate of the use of these services. Additionally, the number of cases of diabetes detected by the information systems can be underestimated in case of early stages of the disease that do not require a drug-based therapy (i.e. diet only treated). Our results, therefore, are conservative, in that the number of people with a new diagnosis of diabetes, detected thanks to the screening programme, is likely to be higher.

In conclusion, we showed that pharmacies could be an appropriate setting for opportunistic screening for type 2 diabetes. Since it has been shown that screening can anticipate the diagnosis of diabetes of 3.3 years on average [4], and that the early identification of diabetes allows to anticipate the correct treatment of the disease and to reduce or delay the onset of severe complications [2], we have reason to believe that the implementation of this community pharmacy-based screening programme can contribute to reduce the negative burden of the disease, with a particular impact on the most vulnerable groups of the population.

Finally, we also believe that there is room to improve the efficiency of the programme through a better selection of participants, so as to make well organised community pharmacy-based screening programmes a useful asset for the national health system.

## Acknowledgments

We thank the Regional Orders of Pharmacists for their essential contribution to the implementation of the project, and Ing. Ezio Festa (ATF Informatics, Cuneo, Italy) for developing the data gathering software and database management. We acknowledge the invaluable work of all the pharmacists who participated in carrying out the programme.

## Author Contributions

**Conceptualization:** Roberto Gnavi, Marco Parente, Massimo Mana, Mario Giaccone, Paolo Cavallo Perin, Giuseppe Costa, Teresa Spadea, Paola Brusa.

**Data curation:** Veronica Sciannameo, Cecilia Scarinzi.

**Formal analysis:** Veronica Sciannameo, Cecilia Scarinzi.

**Funding acquisition:** Giuseppe Costa, Teresa Spadea, Paola Brusa.

**Investigation:** Francesca Baratta.

**Methodology:** Roberto Gnavi, Marco Parente.

**Project administration:** Teresa Spadea, Paola Brusa.

**Software:** Massimo Mana.

**Supervision:** Roberto Gnavi, Paolo Cavallo Perin, Giuseppe Costa, Teresa Spadea.

**Validation:** Cecilia Scarinzi.

**Writing – original draft:** Roberto Gnavi, Veronica Sciannameo, Francesca Baratta, Marco Parente, Teresa Spadea, Paola Brusa.

**Writing – review & editing:** Francesca Baratta, Cecilia Scarinzi, Marco Parente, Mario Giaccone, Paolo Cavallo Perin, Giuseppe Costa, Teresa Spadea, Paola Brusa.

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
