## [Decision Letter · Decision Letter 0]

13 Jan 2020

PONE-D-19-35176

Opportunistic screening for type 2 diabetes in community pharmacies.  Results from a region-wide experience in Italy

PLOS ONE

Dear Dr Gnavi,

Thank you for submitting your manuscript to PLOS ONE. After careful consideration, we feel that it has merit but does not fully meet PLOS ONE’s publication criteria as it currently stands. Therefore, we invite you to submit a revised version of the manuscript that addresses the points raised during the review process.

We would appreciate receiving your revised manuscript by Feb 27 2020 11:59PM. To enhance the reproducibility of your results, we recommend that if applicable you deposit your laboratory protocols in protocols.io, where a protocol can be assigned its own identifier (DOI) such that it can be cited independently in the future. For instructions see: http://journals.plos.org/plosone/s/submission-guidelines#loc-laboratory-protocols

We look forward to receiving your revised manuscript.

Kind regards,

Wen-Jun Tu

Academic Editor

PLOS ONE

Journal Requirements:

2. Thank you for including your ethics statement:  "Ethical Committee Approval Protocol n°46480/13".   

4. Thank you for providing the following Funding Statement: 

"This work was supported by a research grant from the Italian Ministry of Health (www.salute.gov.it) within the Programma CCM 2015: “La farmacia dei servizi per il controllo delle patologie croniche: sperimentazione e trasferimento di un modello di intervento di prevenzione sul diabete di tipo 2”, and an unconditional grant from Federfarma Piemonte (www.federfarma.it).

We note that one or more of the authors is affiliated with the funding organization"Federfarma Piemont", indicating the funder may have had some role in the design, data collection, analysis or preparation of your manuscript for publication; in other words, the funder played an indirect role through the participation of the co-authors.

If the funding organization did not play a role in the study design, data collection and analysis, decision to publish, or preparation of the manuscript and only provided financial support in the form of authors' salaries and/or research materials, please review your statements relating to the author contributions, and ensure you have specifically and accurately indicated the role(s) that these authors had in your study in the Author Contributions section of the online submission form. Please make any necessary amendments directly within this section of the online submission form.  Please also update your Funding Statement to include the following statement: “The funder provided support in the form of salaries for authors [insert relevant initials], but did not have any additional role in the study design, data collection and analysis, decision to publish, or preparation of the manuscript. The specific roles of these authors are articulated in the ‘author contributions’ section.”

If the funding organization did have an additional role, please state and explain that role within your Funding Statement.

Please also provide an updated Competing Interests Statement declaring this commercial affiliation along with any other relevant declarations relating to employment, consultancy, patents, products in development, or marketed products, etc. 

Reviewers' comments:

Reviewer's Responses to Questions

**Comments to the Author**

1. Is the manuscript technically sound, and do the data support the conclusions?

Reviewer #1: Partly

Reviewer #2: Yes

2. Has the statistical analysis been performed appropriately and rigorously? 

Reviewer #1: Yes

Reviewer #2: Yes

3. Have the authors made all data underlying the findings in their manuscript fully available?

Reviewer #1: Yes

Reviewer #2: Yes

4. Is the manuscript presented in an intelligible fashion and written in standard English?

Reviewer #1: Yes

Reviewer #2: Yes

5. Review Comments to the Author

Reviewer #1: The study by Roberto Gnavi et al. aims to evaluate an opportunistic screening strategy based on FINDRISC score for the identification of individuals with type 2 diabetes in Italy.

I would like to applaud the authors impressive efforts of engaging and retain such a large number of pharmacies to voluntarily contribute to the study.

The manuscript is well written and data clearly presented. I have 2 major concerns:

1. The authors mention that no population screening is recommended in Italy. In contrast, North America has several screening guideline recommendations (American Association of Clinical Endocrinologists, American Diabetes Association, Canadian Task Force on Preventive Health Care, and U.S. Preventive Services Task Force) each with their individual set of parameters. What was the authors rationale to use FINDRISC score?

Was there any cost effectiveness analysis performed?

2. The authors mention in the Results section that “pharmacies were randomized to the two arms of the project and 443 were assigned to opportunistic screening”. It is unclear who are the subjects in the control group. The way it is initially described as a cross sectional study and it is not clear at what point randomization occurred. If there was a control group the authors need to describe the results for this group of patients: how many patients were diagnosed with DM in this group? If there was no control group the authors should present data on the pharmacies that were not assigned to the opportunistic screening and then compare it with the intervention group. In this way the validity of the FINDRISC score could be tested in this highly selected group.

Reviewer #2: Gnavi et al, used the FRINDISC questionnaire in community pharmacies to identify patients at risk for diabetes mellitus type 2 (DMT2) in overweight, diabetes free customers aged >45 years. Excluding the number of cases expected during one year follow-up, per 117 interviews one new case of DMT2 was identified. This is an interesting and important study that is easily implemented in daily practice,

Comments:

Results:

Page 9: age is part of the FRINDISC score, so it is evident that higher age is associated with a higher score. Please omit age in this sentence.

Table 1:

First line: either provide the numbers for men or women.

High pressure medication should be replaced by: anti-hypertensive medication.

High glicemic levels, should be replaced by “glycemic”. Please define hyperglycemia in the legenda of this Table.

Define: low physical activity and low vegetable consumption in the legend.

What do the authors mean by “familiarity”?

Table 2

“Glicemia” should be replaced by “glycemic”.

6. PLOS authors have the option to publish the peer review history of their article (what does this mean?). If published, this will include your full peer review and any attached files.

Reviewer #1: No

Reviewer #2: No

---

## [Author Response · Author response to Decision Letter 0]

7 Feb 2020

We included all answers to comments in the "response to reviwers" letter that has been uploaded

---

## [Decision Letter · Decision Letter 1]

18 Feb 2020

Opportunistic screening for type 2 diabetes in community pharmacies.  Results from a region-wide experience in Italy

PONE-D-19-35176R1

Dear Dr. Gnavi,

We are pleased to inform you that your manuscript has been judged scientifically suitable for publication and will be formally accepted for publication once it complies with all outstanding technical requirements.

With kind regards,

Wen-Jun Tu

Academic Editor

PLOS ONE

Additional Editor Comments (optional):

Reviewers' comments:

Reviewer's Responses to Questions

**Comments to the Author**

1. If the authors have adequately addressed your comments raised in a previous round of review and you feel that this manuscript is now acceptable for publication, you may indicate that here to bypass the “Comments to the Author” section, enter your conflict of interest statement in the “Confidential to Editor” section, and submit your "Accept" recommendation.

Reviewer #1: All comments have been addressed

Reviewer #2: All comments have been addressed

2. Is the manuscript technically sound, and do the data support the conclusions?

Reviewer #1: Yes

Reviewer #2: Yes

3. Has the statistical analysis been performed appropriately and rigorously? 

Reviewer #1: Yes

Reviewer #2: Yes

4. Have the authors made all data underlying the findings in their manuscript fully available?

Reviewer #1: Yes

Reviewer #2: Yes

5. Is the manuscript presented in an intelligible fashion and written in standard English?

Reviewer #1: No

Reviewer #2: Yes

6. Review Comments to the Author

Reviewer #1: The authors adequately addressed all the comments. It is an important study that has practical applicability in countries with high diabetes prevalence.

Reviewer #2: I have no further comments fort he authors. The questions are adequately addressed. I have uploaded my review the first time that I reviewed this manuscript.

7. PLOS authors have the option to publish the peer review history of their article (what does this mean?). If published, this will include your full peer review and any attached files.

Reviewer #1: No

Reviewer #2: No

---

## [Editor Report · Acceptance letter]

20 Feb 2020

PONE-D-19-35176R1 

Opportunistic screening for type 2 diabetes in community pharmacies. Results from a region-wide experience in Italy 

Dear Dr. Gnavi:

I am pleased to inform you that your manuscript has been deemed suitable for publication in PLOS ONE. Congratulations! Your manuscript is now with our production department. 

With kind regards,

on behalf of

Dr. Wen-Jun Tu 

Academic Editor

PLOS ONE